# DFNets: Spectral CNNs for Graphs with Feedback-Looped Filters

**Asiri Wijesinghe**
Research School of Computer Science
The Australian National University
`asiri.wijesinghe@anu.edu.au`

**Qing Wang**
Research School of Computer Science
The Australian National University
`qing.wang@anu.edu.au`

## Abstract

We propose a novel spectral convolutional neural network (CNN) model on graph structured data, namely *Distributed Feedback-Looped Networks* (DFNets). This model is incorporated with a robust class of spectral graph filters, called *feedback-looped filters*, to provide better localization on vertices, while still attaining fast convergence and linear memory requirements. Theoretically, feedback-looped filters can guarantee convergence w.r.t. a specified error bound, and be applied universally to any graph without knowing its structure. Furthermore, the propagation rule of this model can diversify features from the preceding layers to produce strong gradient flows. We have evaluated our model using two benchmark tasks: semi-supervised document classification on citation networks and semi-supervised entity classification on a knowledge graph. The experimental results show that our model considerably outperforms the state-of-the-art methods in both benchmark tasks over all datasets.

## 1   Introduction

Convolutional neural networks (CNNs) [20] are a powerful deep learning approach which has been widely applied in various fields, e.g., object recognition [29], image classification [14], and semantic segmentation [22]. Traditionally, CNNs only deal with data that has a regular Euclidean structure, such as images, videos and text. In recent years, due to the rising trends in network analysis and prediction, generalizing CNNs to graphs has attracted considerable interest [3, 7, 11, 26]. However, since graphs are in irregular non-Euclidean domains, this brings up the challenge of how to enhance CNNs for effectively extracting useful features (e.g. topological structure) from arbitrary graphs.

To address this challenge, a number of studies have been devoted to enhancing CNNs by developing filters over graphs. In general, there are two categories of graph filters: (a) spatial graph filters, and (b) spectral graph filters. Spatial graph filters are defined as convolutions directly on graphs, which consider neighbors that are spatially close to a current vertex [1, 9, 11]. In contrast, spectral graph filters are convolutions indirectly defined on graphs, through their spectral representations [3, 5, 7]. In this paper, we follow the line of previous studies in developing spectral graph filters and tackle the problem of designing an effective, yet efficient CNNs with spectral graph filters.

Previously, Bruna et al. [3] proposed convolution operations on graphs via a spectral decomposition of the graph Laplacian. To reduce learning complexity in the setting where the graph structure is not known a priori, Henaff et al. [13] developed a spectral filter with smooth coefficients. Then, Defferrard et al. [7] introduced Chebyshev filters to stabilize convolution operations under coefficient perturbation and these filters can be exactly localized in k-hop neighborhood. Later, Kipf et al. [19] proposed a simple layer-wise propagation model using Chebyshev filters on 1-hop neighborhood. Very recently, some works attempted to develop rational polynomial filters, such as Cayley filters [21]

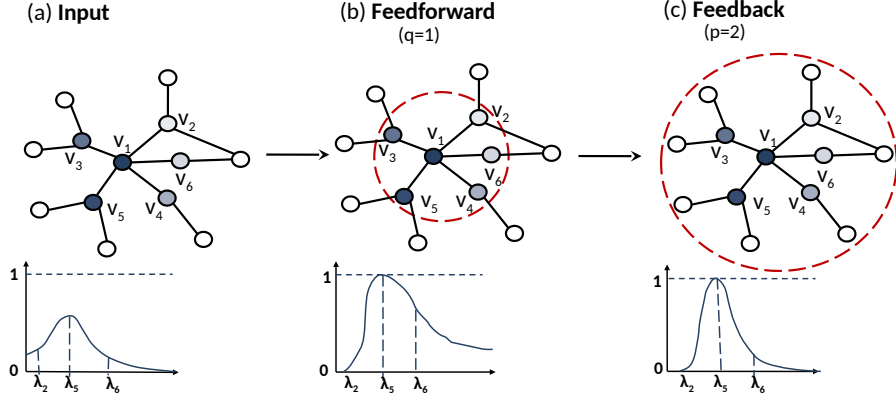

Figure 1: A simplified example of illustrating feedback-looped filters, where $v_1$ is the current vertex and the similarity of the colours indicates the correlation between vertices, e.g., $v_1$ and $v_5$ are highly correlated, but $v_2$ and $v_6$ are less correlated with $v_1$: (a) an input graph, where $\lambda_i$ is the original frequency to vertex $v_i$; (b) the feedforward filtering, which attenuates some low order frequencies, e.g. $\lambda_2$, and amplify other frequencies, e.g. $\lambda_5$ and $\lambda_6$; (c) the feedback filtering, which reduces the error in the frequencies generated by (b), e.g. $\lambda_6$.

and ARMA$_1$ [2]. From a different perspective, Petar et al. [31] proposed a self-attention based CNN architecture for graph filters, which extracts features by considering the importance of neighbors.

One key idea behind existing works on designing spectral graph filters is to approximate the frequency responses of graph filters using a polynomial function (e.g. Chebyshev filters [7]) or a rational polynomial function (e.g. Cayley filters [21] and ARMA$_1$ [2]). Polynomial filters are sensitive to changes in the underlying graph structure. They are also very smooth and can hardly model sharp changes, as illustrated in Figure 1. Rational polynomial filters are more powerful to model localization, but they often have to trade off computational efficiency, resulting in higher learning and computational complexities, as well as instability.

**Contributions.** In this work, we aim to develop a new class of spectral graph filters that can overcome the above limitations. We also propose a spectral CNN architecture (i.e. DFNet) to incorporate these graph filters. In summary, our contributions are as follows:

- **Improved localization.** A new class of spectral graph filters, called *feedback-looped filters*, is proposed to enable better localization, due to its rational polynomial form. Basically, feedback-looped filters consist of two parts: *feedforward* and *feedback*. The feedforward filtering is k-localized as polynomial filters, while the feedback filtering is unique which refines k-localized features captured by the feedforward filtering to improve approximation accuracy. We also propose two techniques: *scaled-normalization* and *cut-off frequency* to avoid the issues of gradient vanishing/exploding and instabilities.

- **Efficient computation.** For feedback-looped filters, we avoid the matrix inversion implied by the denominator through approximating the matrix inversion with a recursion. Thus, benefited from this approximation, feedback-looped filters attain linear convergence time and linear memory requirements w.r.t. the number of edges in a graph.

- **Theoretical properties.** Feedback-looped filters enjoy several nice theoretical properties. Unlike other rational polynomial filters for graphs, they have theoretically guaranteed convergence w.r.t. a specified error bound. On the other hand, they still have the universal property as other spectral graph filters [17], i.e., can be applied without knowing the underlying structure of a graph. The optimal coefficients of feedback-looped filters are learnable via an optimization condition for any given graph.

- **Dense architecture.** We propose a layer-wise propagation rule for our spectral CNN model with feedback-looped filters, which densely connects layers as in DenseNet [15]. This design enables our model to diversify features from all preceding layers, leading to a strong gradient flow. We also introduce a layer-wise regularization term to alleviate the overfitting issue. In doing so, we can prevent the generation of spurious features and thus improve accuracy of the prediction.

To empirically verify the effectiveness of our work, we have evaluated feedback-looped filters within three different CNN architectures over four benchmark datasets to compare against the state-of-the-art methods. The experimental results show that our models significantly outperform the state-of-the-art methods. We further demonstrate the effectiveness of our model DFNet through the node embeddings in a 2-D space of vertices from two datasets.

## 2 Spectral Convolution on Graphs

Let $G = (V, E, A)$ be an undirected and weighted graph, where $V$ is a set of vertices, $E \subseteq V \times V$ is a set of edges, and $A \in \mathbb{R}^{n \times n}$ is an adjacency matrix which encodes the weights of edges. We let $n = |V|$ and $m = |E|$. A *graph signal* is a function $x : V \rightarrow \mathbb{R}$ and can be represented as a vector $x \in \mathbb{R}^n$ whose $i^{th}$ component $x_i$ is the value of $x$ at the $i^{th}$ vertex in $V$. The graph Laplacian is defined as $L = I - D^{-1/2}AD^{-1/2}$, where $D \in \mathbb{R}^{n \times n}$ is a diagonal matrix with $D_{ii} = \sum_j A_{ij}$ and $I$ is an identity matrix. $L$ has a set of orthogonal eigenvectors $\{u_i\}_{i=0}^{n-1} \in \mathbb{R}^n$, known as the *graph Fourier basis*, and non-negative eigenvalues $\{\lambda_i\}_{i=0}^{n-1}$, known as the *graph frequencies* [5]. $L$ is diagonalizable by the eigendecomposition such that $L = U\Lambda U^H$, where $\Lambda = diag\left([\lambda_0, \ldots, \lambda_{n-1}]\right) \in \mathbb{R}^{n \times n}$ and $U^H$ is a hermitian transpose of $U$. We use $\lambda_{min}$ and $\lambda_{max}$ to denote the smallest and largest eigenvalues of $L$, respectively.

Given a graph signal $x$, the *graph Fourier transform* of $x$ is $\hat{x} = U^H x \in \mathbb{R}^n$ and its inverse is $x = U\hat{x}$ [27, 30]. The graph Fourier transform enables us to apply graph filters in the vertex domain. A *graph filter* $h$ can filter $x$ by altering (amplifying or attenuating) the graph frequencies as

$$h(L)x = h(U\Lambda U^H)x = Uh(\Lambda)U^H x = Uh(\Lambda)\hat{x}. \tag{1}$$

Here, $h(\Lambda) = diag([h(\lambda_0), \ldots, h(\lambda_{n-1})])$, which controls how the frequency of each component in a graph signal $x$ is modified. However, applying graph filtering as in Eq. 1 requires the eigendecomposition of $L$, which is computationally expensive. To address this issue, several works [2, 7, 12, 19, 21, 23] have studied the approximation of $h(\Lambda)$ by a polynomial or rational polynomial function.

**Chebyshev filters.** Hammond et al. [12] first proposed to approximate $h(\lambda)$ by a polynomial function with $k^{th}$-order polynomials and Chebyshev coefficients. Later, Defferrard et al. [7] developed Chebyshev filters for spectral CNNs on graphs. A Chebyshev filter is defined as

$$h_\theta(\tilde{\lambda}) = \sum_{j=0}^{k-1} \theta_j T_j(\tilde{\lambda}), \tag{2}$$

where $\theta \in \mathbb{R}^k$ is a vector of learnable Chebyshev coefficients, $\tilde{\lambda} \in [-1, 1]$ is rescaled from $\lambda$, the Chebyshev polynomials $T_j(\lambda) = 2\lambda T_{j-1}(\lambda) - T_{j-2}(\lambda)$ are recursively defined with $T_0(\lambda) = 1$ and $T_1(\lambda) = \lambda$, and $k$ controls the size of filters, i.e., localized in k-hop neighborhood of a vertex [12]. Kipf and Welling [19] simplified Chebyshev filters by restricting to 1-hop neighborhood.

**Lanczos filters.** Recently, Liao et al. [23] used the Lanczos algorithm to generate a low-rank matrix approximation $T$ for the graph Laplacian. They used the affinity matrix $S = D^{-1/2}AD^{-1/2}$. Since $L = I - S$ holds, $L$ and $S$ share the same eigenvectors but have different eigenvalues. As a result, $L$ and $S$ correspond to the same $\hat{x}$. To approximate the eigenvectors and eigenvalues of $S$, they diagonalize the tri-diagonal matrix $T \in \mathbb{R}^{m \times m}$ to compute Ritz-vectors $V \in \mathbb{R}^{n \times m}$ and Ritz-values $R \in \mathbb{R}^{m \times m}$, and thus $S \approx VRV^T$. Accordingly, a k-hop Lanczos filter operation is,

$$h_\theta(R) = \sum_{j=0}^{k-1} \theta_j R^j, \tag{3}$$

where $\theta \in \mathbb{R}^k$ is a vector of learnable Lanczos filter coefficients. Thus, spectral convolutional operation is defined as $h_\theta(S)x \approx Vh_\theta(R)V^T x$. Such Lanczos filter operations can significantly reduce computation overhead when approximating large powers of $S$, i.e. $S^k \approx VR^kV^T$. Thus, they can efficiently compute the spectral graph convolution with a very large localization range to easily capture the multi-scale information of the graph.

**Cayley filters.** Observing that Chebyshev filters have difficulty in detecting narrow frequency bands due to $\tilde{\lambda} \in [-1, 1]$, Levie et al. [21] proposed Cayley filters, based on Cayley polynomials:

$$h_{\theta,s}(\lambda) = \theta_0 + 2Re(\sum_{j=1}^{k-1} \theta_j(s\lambda - i)^j(s\lambda + i)^{-j}),\tag{4}$$

where $\theta_0 \in \mathbb{R}$ is a real coefficient and $(\theta_1, \ldots, \theta_{k-1}) \in \mathbb{C}^{k-1}$ is a vector of complex coefficients. $Re(x)$ denotes the real part of a complex number $x$, and $s > 0$ is a parameter called *spectral zoom*, which controls the degree of "zooming" into eigenvalues in $\Lambda$. Both $\theta$ and $s$ are learnable during training. To improve efficiency, the Jacobi method is used to approximately compute Cayley polynomials.

**ARMA$_1$ filters.** Bianchi et al. [2] sought to address similar issues as identified in [21]. However, different from Cayley filters, they developed a first-order ARMA filter, which is approximated by a first-order recursion:

$$\bar{x}^{(t+1)} = a\tilde{L}\bar{x}^{(t)} + bx,\tag{5}$$

where $a$ and $b$ are the filter coefficients, $\bar{x}^{(0)} = x$, and $\tilde{L} = (\lambda_{max} - \lambda_{min})/2I - L$. Accordingly, the frequency response is defined as:

$$h(\tilde{\lambda}) = \frac{r}{\tilde{\lambda} - p},\tag{6}$$

where $\tilde{\lambda} = (\lambda_{max} - \lambda_{min})/2\lambda$, $r = -b/a$, and $p = 1/a$ [17]. Multiple ARMA$_1$ filters can be applied in parallel to obtain a ARMA$_k$ filter. However, the memory complexity of $k$ parallel ARMA$_1$ filters is $k$ times higher than ARMA$_1$ graph filters.

We make some remarks on how these existing spectral filters are related to each other. (i) As discussed in [2, 21, 23], polynomial filters (e.g. Chebyshev and Lanczos filters) can be approximately treated as a special kind of rational polynomial filters. (ii) Further, Chebyshev filters can be regarded as a special case of Lanczos filters. (iii) Although both Cayley and ARMA$_k$ filters are rational polynomial filters, they differ in how they approximate the matrix inverse implied by the denominator of a rational function. Cayley filters use a fixed number of Jacobi iterations, while ARMA$_k$ filters use a first-order recursion plus a parallel bank of $k$ ARMA$_1$. (iv) ARMA$_1$ by Bianchi et al. [2] is similar to GCN by Kipf et al. [19] because they both consider localization within 1-hop neighborhood.

## 3 Proposed Method

We introduce a new class of spectral graph filters, called *feedback-looped filters*, and propose a spectral CNN for graphs with feedback-looped filters, namely *Distributed Feedback-Looped Networks* (DFNets). We also discuss optimization techniques and analyze theoretical properties.

### 3.1 Feedback-Looped Filters

Feedback-looped filters belong to a class of Auto Regressive Moving Average (ARMA) filters [16, 17]. Formally, an ARMA filter is defined as:

$$h_{\psi,\phi}(L)x = \Big(I + \sum_{j=1}^{p} \psi_j L^j\Big)^{-1}\Big(\sum_{j=0}^{q} \phi_j L^j\Big)x.\tag{7}$$

The parameters $p$ and $q$ refer to the *feedback* and *feedforward* degrees, respectively. $\psi \in \mathbb{C}^p$ and $\phi \in \mathbb{C}^{q+1}$ are two vectors of complex coefficients. Computing the denominator of Eq. 7 however requires a matrix inversion, which is computationally inefficient for large graphs. To circumvent this issue, *feedback-looped filters* use the following approximation:

$$\bar{x}^{(0)} = x \text{ and } \bar{x}^{(t)} = -\sum_{j=1}^{p} \psi_j \tilde{L}^j \bar{x}^{(t-1)} + \sum_{j=0}^{q} \phi_j \tilde{L}^j x,\tag{8}$$

where $\tilde{L} = \hat{L} - (\frac{\hat{\lambda}_{max}}{2})I$, $\hat{L} = I - \hat{D}^{-1/2}\hat{A}\hat{D}^{-1/2}$, $\hat{A} = A + I$, $\hat{D}_{ii} = \sum_j \hat{A}_{ij}$ and $\hat{\lambda}_{max}$ is the largest eigenvalue of $\hat{L}$. Accordingly, the frequency response of feedback-looped filters is defined as:

$$h(\lambda_i) = \frac{\sum_{j=0}^{q} \phi_j \lambda_i^j}{1 + \sum_{j=1}^{p} \psi_j \lambda_i^j}.\tag{9}$$

To alleviate the issues of gradient vanishing/exploding and numerical instabilities, we further introduce two techniques in the design of feedback-looped filters: *scaled-normalization* and *cut-off frequency*.

**Scaled-normalization technique.** To assure the stability of feedback-looped filters, we apply the scaled-normalization technique to increasing the stability region, i.e., using the scaled-normalized Laplacian $\tilde{L} = \hat{L} - (\frac{\hat{\lambda}_{max}}{2})I$, rather than just $\hat{L}$. This accordingly helps centralize the eigenvalues of the Laplacian $\hat{L}$ and reduce its spectral radius bound. The scaled-normalized Laplacian $\tilde{L}$ consists of graph frequencies within $[0, 2]$, in which eigenvalues are ordered in an increasing order.

**Cut-off frequency technique.** To map graph frequencies within $[0, 2]$ to a uniform discrete distribution, we define a *cut-off frequency* $\lambda_{cut} = (\frac{\lambda_{max}}{2} - \eta)$, where $\eta \in [0, 1]$ and $\lambda_{max}$ refers to the largest eigenvalue of $\tilde{L}$. The cut-off frequency is used as a threshold to control the amount of attenuation on graph frequencies. The eigenvalues $\{\lambda_i\}_{i=0}^{n-1}$ are converted to binary values $\{\tilde{\lambda}_i\}_{i=0}^{n-1}$ such that $\tilde{\lambda}_i = 1$ if $\lambda_i \geq \lambda_{cut}$ and $\tilde{\lambda}_i = 0$ otherwise. This trick allows the generation of ideal high-pass filters so as to sharpen a signal by amplifying its graph Fourier coefficients. This technique also solves the issue of narrow frequency bands existing in previous spectral filters, including both polynomial and rational polynomial filters [7, 21]. This is because these previous spectral filters only accept a small band of frequencies. In contrast, our proposed feedback-looped filters resolve this issue using a cut-off frequency technique, i.e., amplifying frequencies higher than a certain low cut-off value while attenuating frequencies lower than that cut-off value. Thus, our proposed filters can accept a wider range of frequencies and capture better characteristic properties of a graph.

## 3.2 Coefficient Optimisation

Given a feedback-looped filter with a desired frequency response: $\hat{h} : \{\tilde{\lambda}_i\}_{i=0}^{n-1} \to \mathbb{R}$, we aim to find the optimal coefficients $\psi$ and $\phi$ that make the frequency response as close as possible to the desired frequency response, i.e. to minimize the following error:

$$\acute{e}(\tilde{\lambda}_i) = \hat{h}(\tilde{\lambda}_i) - \frac{\sum_{j=0}^{q} \phi_j \tilde{\lambda}_i^j}{1 + \sum_{j=1}^{p} \psi_j \tilde{\lambda}_i^j} \tag{10}$$

However, the above equation is not linear w.r.t. the coefficients $\psi$ and $\phi$. Thus, we redefine the error as follows:

$$e(\tilde{\lambda}_i) = \hat{h}(\tilde{\lambda}_i) + \hat{h}(\tilde{\lambda}_i) \sum_{j=1}^{p} \psi_j \tilde{\lambda}_i^j - \sum_{j=0}^{q} \phi_j \tilde{\lambda}_i^j. \tag{11}$$

Let $e = [e(\tilde{\lambda}_0), \ldots, e(\tilde{\lambda}_{n-1})]^T$, $\hat{h} = [\hat{h}(\tilde{\lambda}_0), \ldots, \hat{h}(\tilde{\lambda}_{n-1})]^T$, $\alpha \in \mathbb{R}^{n \times p}$ with $\alpha_{ij} = \tilde{\lambda}_i^j$ and $\beta \in \mathbb{R}^{n \times (q+1)}$ with $\beta_{ij} = \tilde{\lambda}_i^{j-1}$ are two Vandermonde-like matrices. Then, we have $e = \hat{h} + diag(\hat{h})\alpha\psi - \beta\phi$. Thus, the stable coefficients $\psi$ and $\phi$ can be learned by minimizing $e$ as a convex constrained least-squares optimization problem:

$$\mathbf{minimize}_{\psi,\phi} \ ||\hat{h} + diag(\hat{h})\alpha\psi - \beta\phi||_2 \tag{12}$$

$$\mathbf{subject\ to} \ ||\alpha\psi||_\infty \leq \gamma \text{ and } \gamma < 1$$

Here, the parameter $\gamma$ controls the tradeoff between convergence efficiency and approximation accuracy. A higher value of $\gamma$ can lead to slower convergence but better accuracy. It is not recommended to have very low $\gamma$ values due to potentially unacceptable accuracy. $||\alpha\psi||_\infty \leq \gamma < 1$ is the stability condition, which will be further discussed in detail in Section 3.4.

## 3.3 Spectral Convolutional Layer

We propose a CNN-based architecture, called DFNets, which can stack multiple spectral convolutional layers with feedback-looped filters to extract features of increasing abstraction. Let $\mathbf{P} = -\sum_{j=1}^{p} \psi_j \tilde{L}^j$ and $\mathbf{Q} = \sum_{j=0}^{q} \phi_j \tilde{L}^j$. The propagation rule of a spectral convolutional layer is defined as:

$$\bar{X}^{(t+1)} = \sigma(\mathbf{P}\bar{X}^{(t)}\theta_1^{(t)} + \mathbf{Q}X\theta_2^{(t)} + \mu(\theta_1^{(t)}; \theta_2^{(t)}) + b), \tag{13}$$

where $\sigma$ refers to a non-linear activation function such as $ReLU$. $\bar{X}^{(0)} = X \in \mathbb{R}^{n \times f}$ is a graph signal matrix where $f$ refers to the number of features. $\bar{X}^{(t)}$ is a matrix of activations in the $t^{th}$ layer. $\theta_1^{(t)} \in \mathbb{R}^{c \times h}$ and $\theta_2^{(t)} \in \mathbb{R}^{f \times h}$ are two trainable weight matrices in the $t^{th}$ layer. To compute $\bar{X}^{(t+1)}$, a vertex needs access to its $p$-hop neighbors with the output signal of the previous layer $\bar{X}^{(t)}$, and its $q$-hop neighbors with the input signal from $X$. To attenuate the overfitting issue, we add $\mu(\theta_1^{(t)}; \theta_2^{(t)})$, namely *kernel regularization* [6], and a bias term $b$. We use the xavier normal initialization method [10] to initialise the kernel and bias weights, the unit-norm constraint technique [8] to normalise the kernel and bias weights by restricting parameters of all layers in a small range, and the kernel regularization technique to penalize the parameters in each layer during the training. In doing so, we can prevent the generation of spurious features and thus improve the accuracy of prediction [1].

In this model, each layer is directly connected to all subsequent layers in a feed-forward manner, as in DenseNet [15]. Consequently, the $t^{th}$ layer receives all preceding feature maps $F_0, \ldots, F_{t-1}$ as input. We concatenate multiple preceding feature maps column-wise into a single tensor to obtain more diversified features for boosting the accuracy. This densely connected CNN architecture has several compelling benefits: (a) reduce the vanishing-gradient issue, (b) increase feature propagation and reuse, and (c) refine information flow between layers [15].

### 3.4 Theoretical Analysis

Feedback-looped filters have several nice properties, e.g., guaranteed convergence, linear convergence time, and universal design. We discuss these properties and analyze computational complexities.

**Convergence.** Theoretically, a feedback-looped filter can achieve a desired frequency response only when $t \to \infty$ [17]. However, due to the property of linear convergence preserved by feedback-looped filters, stability can be guaranteed after a number of iterations w.r.t. a specified small error [16]. More specifically, since the pole of rational polynomial filters should be in the unit circle of the z-plane to guarantee the stability, we can derive the stability condition $|| - \sum_{j=1}^{p} \psi_j L^j || < 1$ by Eq. 7 in the vertex domain and correspondingly obtain the stability condition $||\alpha\psi||_\infty \leq \gamma \in (0, 1)$ in the frequency domain as stipulated in Eq. 12 [16].

**Universal design.** The universal design is beneficial when the underlying structure of a graph is unknown or the topology of a graph changes over time. The corresponding filter coefficients can be learned independently of the underlying graph and are universally applicable. When designing feedback-looped filters, we define the desired frequency response function $\hat{h}$ over graph frequencies $\tilde{\lambda}_i$ in a binary format in the uniform discrete distribution as discussed in Section 3.1. Then, we solve Eq. 12 in the least-squares sense for this finite set of graph frequencies to find optimal filter coefficients.

| Spectral Graph Filter | Type | Learning Complexity | Time Complexity | Memory Complexity |
|---|---|---|---|---|
| Chebyshev filters [7] | Polynomial | $O(k)$ | $O(km)$ | $O(m)$ |
| Lanczos filters [23] | | $O(k)$ | $O(km^2)$ | $O(m^2)$ |
| Cayley filters [21] | Rational polynomial | $O((r+1)k)$ | $O((r+1)km)$ | $O(m)$ |
| ARMA$_1$ filters [2] | | $O(t)$ | $O(tm)$ | $O(m)$ |
| $d$ parallel ARMA$_1$ filters [2] | | $O(t)$ | $O(tm)$ | $O(dm)$ |
| Feedback-looped filters (ours) | | $O(tp+q)$ | $O((tp+q)m)$ | $O(m)$ |

Table 1: Learning, time and space complexities of spectral graph filters.

**Complexity.** When computing $\bar{x}^{(t)}$ as in Eq. 8, we need to calculate $\tilde{L}^j \bar{x}^{(t-1)}$ for $j = 1, \ldots, p$ and $\tilde{L}^j x$ for $j = 1, \ldots, q$. Nevertheless, $\tilde{L}^j x$ is computed only once because $\tilde{L}^j x = \tilde{L}(\tilde{L}^{j-1} x)$. Thus, we need $p$ multiplications for each $t$ in the first term in Eq. 8, and $q$ multiplications for the second term in Eq. 8. Table 1 summarizes the complexity results of existing spectral graph filters and ours, where $r$ refers to the number of Jacobi iterations in [21]. Note that, when $t = 1$ (i.e., one spectral convolutional layer), feedback-looped filters have the same learning, time and memory complexities as Chebyshev filters, where $p + q = k$.

# 4 Numerical Experiments

We evaluate our models on two benchmark tasks: (1) semi-supervised document classification in citation networks, and (2) semi-supervised entity classification in a knowledge graph.

## 4.1 Experimental Set-Up

**Datasets.** We use three citation network datasets Cora, Citeseer, and Pubmed [28] for semi-supervised document classification, and one dataset NELL [4] for semi-supervised entity classification. NELL is a bipartite graph extracted from a knowledge graph [4]. Table 2 contains dataset statistics [33].

| Dataset | Type | #Nodes | #Edges | #Classes | #Features | %Labeled Nodes |
|---------|------|--------|--------|----------|-----------|----------------|
| Cora | Citation network | 2,708 | 5,429 | 7 | 1,433 | 0.052 |
| Citeseer | Citation network | 3,327 | 4,732 | 6 | 3,703 | 0.036 |
| Pubmed | Citation network | 19,717 | 44,338 | 3 | 500 | 0.003 |
| NELL | Knowledge graph | 65,755 | 266,144 | 210 | 5,414 | 0.001 |

Table 2: Dataset statistics.

**Baseline methods.** We compare against twelve baseline methods, including five methods using spatial graph filters, i.e., Semi-supervised Embedding (SemiEmb) [32], Label Propagation (LP) [34], skip-gram graph embedding model (DeepWalk) [26], Iterative Classification Algorithm (ICA) [24], and semi-supervised learning with graph embedding (Planetoid*) [33], and seven methods using spectral graph filters: Chebyshev [7], Graph Convolutional Networks (GCN) [19], Lanczos Networks (LNet) and Adaptive Lanczos Networks (AdaLNet) [23], CayleyNet [21], Graph Attention Networks (GAT) [31], and ARMA Convolutional Networks (ARMA$_1$) [2].

We evaluate our feedback-looped filters using three different spectral CNN models: (i) DFNet: a densely connected spectral CNN with feedback-looped filters, (ii) DFNet-ATT: a self-attention based densely connected spectral CNN with feedback-looped filters, and (iii) DF-ATT: a self-attention based spectral CNN model with feedback-looped filters.

| Model | L2 reg. | #Layers | #Units | Dropout | [p, q] | $\lambda_{cut}$ |
|-------|---------|---------|--------|---------|--------|-----------------|
| DFNet | 9e-2 | 5 | [8, 16, 32, 64, 128] | 0.9 | [5, 3] | 0.5 |
| DFNet-ATT | 9e-4 | 4 | [8, 16, 32, 64] | 0.9 | [5, 3] | 0.5 |
| DF-ATT | 9e-3 | 2 | [32, 64] | [0.1, 0.9] | [5, 3] | 0.5 |

Table 3: Hyperparameter settings for citation network datasets.

**Hyperparameter settings.** We use the same data splitting for each dataset as in Yang et al. [33]. The hyperparameters of our models are initially selected by applying the orthogonalization technique (a randomized search strategy). We also use a layerwise regularization (L2 regularization) and bias terms to attenuate the overfitting issue. All models are trained 200 epochs using the Adam optimizer [18] with a learning rate of 0.002. Table 3 summarizes the hyperparameter settings for citation network datasets. The same hyperparameters are applied to the NELL dataset except for L2 regularization (i.e., 9e-2 for DFNet and DFnet-ATT, and 9e-4 for DF-ATT). For $\gamma$, we choose the best setting for each model. For self-attention, we use 8 multi-attention heads and 0.5 attention dropout for DFNet-ATT, and 6 multi-attention heads and 0.3 attention dropout for DF-ATT. The parameters $p = 5$, $q = 3$ and $\lambda_{cut} = 0.5$ are applied to all three models over all datasets.

## 4.2 Comparison with Baseline Methods

Table 4 summarizes the results of classification in terms of accuracy. The results of the baseline methods are taken from the previous works [19, 23, 31, 33]. Our models DFNet and DFNet-ATT outperform all the baseline methods over four datasets. Particularly, we can see that: (1) Compared with polynomial filters, DFNet improves upon GCN (which performs best among the models using polynomial filters) by a margin of 3.7%, 3.9%, 5.3% and 2.3% on the datasets Cora, Citeseer, Pubmed and NELL, respectively. (2) Compared with rational polynomial filters, DFNet improves upon CayleyNet and ARMA$_1$ by 3.3% and 1.8% on the Cora dataset, respectively. For the other datasets, CayleyNet does not have results available in [21]. (3) DFNet-ATT further improves the results of DFNet due to the addition of a self-attention layer. (4) Compared with GAT (Chebyshev filters with

self-attention), DF-ATT also improves the results and achieves 0.4%, 0.6% and 3.3% higher accuracy on the datasets Cora, Citeseer and Pubmed, respectively.

Additionally, we compare DFNet (our feedback-looped filters + DenseBlock) with GCN + Dense-Block and GAT + DenseBlock. The results are also presented in Table 4. We can see that our feedback-looped filters perform best, no matter whether or not the dense architecture is used.

| Model | Cora | Citeseer | Pubmed | NELL |
|---|---|---|---|---|
| SemiEmb [32] | 59.0 | 59.6 | 71.1 | 26.7 |
| LP [34] | 68.0 | 45.3 | 63.0 | 26.5 |
| DeepWalk [26] | 67.2 | 43.2 | 65.3 | 58.1 |
| ICA [24] | 75.1 | 69.1 | 73.9 | 23.1 |
| Planetoid* [33] | 64.7 | 75.7 | 77.2 | 61.9 |
| Chebyshev [7] | 81.2 | 69.8 | 74.4 | - |
| GCN [19] | 81.5 | 70.3 | 79.0 | 66.0 |
| LNet [23] | 79.5 | 66.2 | 78.3 | - |
| AdaLNet [23] | 80.4 | 68.7 | 78.1 | - |
| CayleyNet [21] | 81.9* | - | - | - |
| $ARMA_1$ [2] | 83.4 | 72.5 | 78.9 | - |
| GAT [31] | 83.0 | 72.5 | 79.0 | - |
| GCN + DenseBlock | $82.7 \pm 0.5$ | $71.3 \pm 0.3$ | $81.5 \pm 0.5$ | $66.4 \pm 0.3$ |
| GAT + Dense Block | $83.8 \pm 0.3$ | $73.1 \pm 0.3$ | $81.8 \pm 0.3$ | - |
| DFNet (ours) | $\mathbf{85.2 \pm 0.5}$ | $\mathbf{74.2 \pm 0.3}$ | $\mathbf{84.3 \pm 0.4}$ | $\mathbf{68.3 \pm 0.4}$ |
| DFNet-ATT (ours) | $\mathbf{86.0 \pm 0.4}$ | $\mathbf{74.7 \pm 0.4}$ | $\mathbf{85.2 \pm 0.3}$ | $\mathbf{68.8 \pm 0.3}$ |
| DF-ATT (ours) | $83.4 \pm 0.5$ | $73.1 \pm 0.4$ | $\mathbf{82.3 \pm 0.3}$ | $\mathbf{67.6 \pm 0.3}$ |

Table 4: Accuracy (%) averaged over 10 runs (* was obtained using a different data splitting in [21])

## 4.3 Comparison under Different Polynomial Orders

In order to test how the polynomial orders $p$ and $q$ influence the performance of our model DFNet, we conduct experiments to evaluate DFNet on three citation network datasets using different polynomial orders $p = [1, 3, 5, 7, 9]$ and $q = [1, 3, 5, 7, 9]$. Figure 2 presents the experimental results. In our experiments, $p = 5$ and $q = 3$ turn out to be the best parameters for DFNet over these datasets. In other words, this means that feedback-looped filters are more stable on $p = 5$ and $q = 3$ than other values of $p$ and $q$. This is because, when $p = 5$ and $q = 3$, Eq. 12 can obtain better convergence for finding optimal coefficients than in the other cases. Furthermore, we observe that: (1) Setting $p$ to be too low or too high can both lead to poor performance, as shown in Figure 2.(a), and (2) when $q$ is larger than $p$, the accuracy decreases rapidly as shown in Figure 2.(b). Thus, when choosing $p$ and $q$, we require that $p > q$ holds.

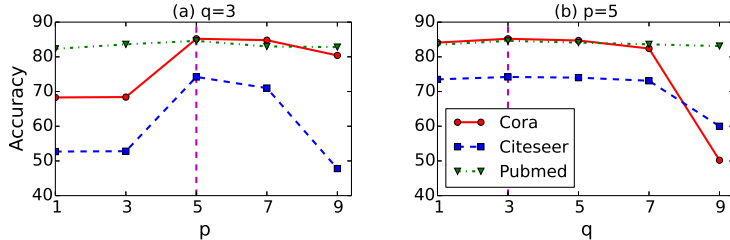

Figure 2: Accuracy (%) of DFNet under different polynomial orders $p$ and $q$.

## 4.4 Evaluation of Scaled-Normalization and Cut-off Frequency

To understand how effectively the scaled-normalisation and cut-off frequency techniques can help learn graph representations, we compare our methods that implement these techniques with the variants of our methods that only implement one of these techniques. The results are presented in Figure 3. We can see that, the models using these two techniques outperform the models that only use one of these techniques over all citation network datasets. Particularly, the improvement is significant on the Cora and Citeseer datasets.

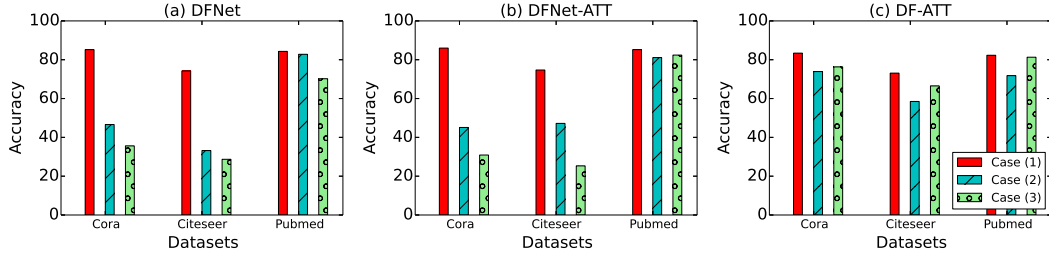

Figure 3: Accuracy (%) of our models in three cases: (1) using both scaled-normalization and cut-off frequency, (2) using only cut-off frequency, and (3) using only scaled-normalization.

## 4.5 Node Embeddings

We analyze the node embeddings by DFNets over two datasets: Cora and Pubmed in a 2-D space. Figures 4 and 5 display the visualization of the learned 2-D embeddings of GCN, GAT, and DFNet (ours) on Pubmed and Cora citation networks by applying t-SNE [25] respectively. Colors denote different classes in these datasets. It reveals the clustering quality of theses models. These figures clearly show that our model DFNet has better separated 3 and 7 clusters respectively in the embedding spaces of Pubmed and Cora datasets. This is because features extracted by DFNet yield better node representations than GCN and GAT models.

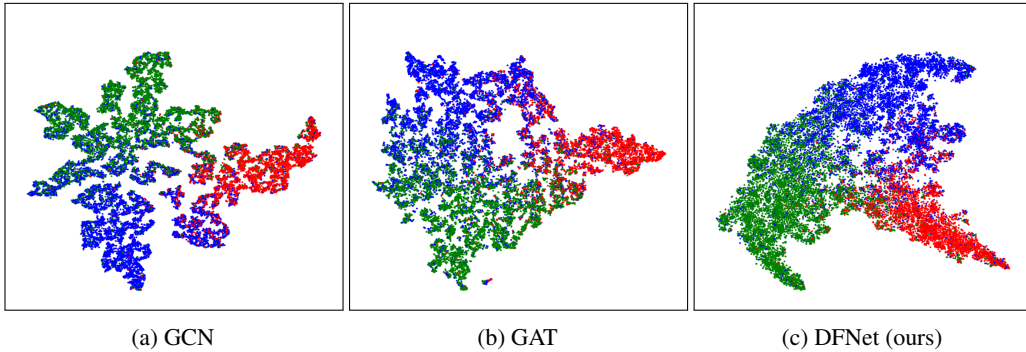

(a) GCN        (b) GAT        (c) DFNet (ours)

Figure 4: The t-SNE visualization of the 2-D node embedding space for the Pubmed dataset.

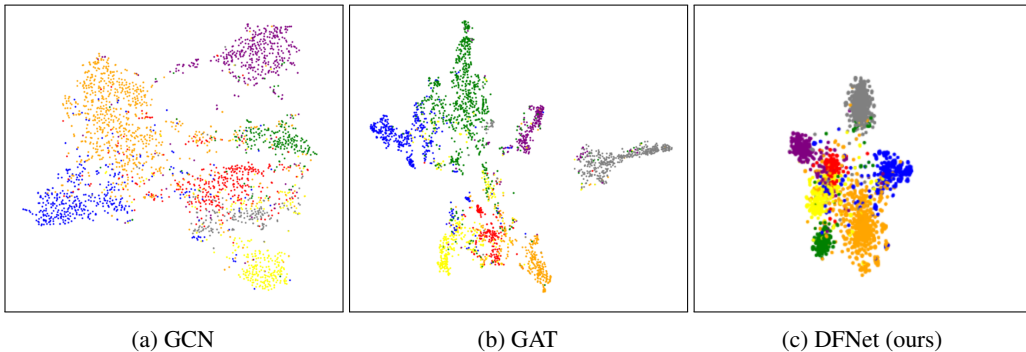

(a) GCN        (b) GAT        (c) DFNet (ours)

Figure 5: The t-SNE visualization of the 2-D node embedding space for the Cora dataset.

## 5 Conclusions

In this paper, we have introduced a spectral CNN architecture (DFNets) with feedback-looped filters on graphs. To improve approximation accuracy, we have developed two techniques: scaled normalization and cut-off frequency. In addition to these, we have discussed some nice properties of feedback-looped filters, such as guaranteed convergence, linear convergence time, and universal design. Our proposed model outperforms the state-of-the-art approaches significantly in two benchmark tasks. In future, we plan to extend the current work to time-varying graph structures. As discussed in [17], feedback-looped graph filters are practically appealing for time-varying settings, and similar to static graphs, some nice properties would likely hold for graphs that are a function of time.

## Footnotes

[1] DFNets implementation can be found at: https://github.com/wokas36/DFNets

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

# Appendices

In the following, we provide further experiments on comparing our work with the others.

**Comparison with different spectral graph filters.** We have conducted an ablation study of our proposed graph filters. Specifically, we compare our feedback-looped filters, i.e., the newly proposed spectral filters in this paper, against other spectral filters such as Chebyshev filters and Cayley filters. To conduct this ablation study, we remove the dense connections from our model DFNet. The experimental results are presented in table 5. It shows that feedback-looped filters improve localization upon Chebyshev filters by a margin of 1.4%, 1.7% and 7.3% on the datasets Cora, Citeseer and Pubmed, respectively. It also improves upon Cayley filters by a margin of 0.7% on the Cora dataset.

| Model | Cora | Citeseer | Pubmed |
|---|---|---|---|
| Chebyshev filters [7] | 81.2 | 69.8 | 74.4 |
| Cayley filters [21] | 81.9 | - | - |
| Feedback-looped filters (ours) | **82.6 ± 0.3** | **71.5 ± 0.4** | **81.7 ± 0.6** |

Table 5: Accuracy (%) averaged over 10 runs.

**Comparison with LNet and AdaLNet using different data splittings.** We have benchmarked the performance of our DFNet model against the models LNet and AdaLNet proposed in [23], as well as Chebyshev, GCN and GAT, over three citation network datasets Cora, Citeseer and Pubmed. We use the same data splittings as used in [23]. All the experiments are repeated 10 times. For our model DFNet, we use the same hyperparameter settings as discussed in Section 4.2.

| Training Split | Chebyshev | GCN | GAT | LNet | AdaLNet | DFNet |
|---|---|---|---|---|---|---|
| 5.2% (standard) | 78.0 ± 1.2 | 80.5 ± 0.8 | 82.6 ± 0.7 | 79.5 ± 1.8 | 80.4 ± 1.1 | **85.2 ± 0.5** |
| 3% | 62.1 ± 6.7 | 74.0 ± 2.8 | 56.8 ± 7.9 | 76.3 ± 2.3 | 77.7 ± 2.4 | **80.5 ± 0.4** |
| 1% | 44.2 ± 5.6 | 61.0 ± 7.2 | 48.6 ± 8.0 | 66.1 ± 8.2 | 67.5 ± 8.7 | **69.5 ± 2.3** |
| 0.5% | 33.9 ± 5.0 | 52.9 ± 7.4 | 41.4 ± 6.9 | 58.1 ± 8.2 | 60.8 ± 9.0 | **61.3 ± 4.3** |

Table 6: Accuracy (%) averaged over 10 runs on the Cora dataset.

| Training Split | Chebyshev | GCN | GAT | LNet | AdaLNet | DFNet |
|---|---|---|---|---|---|---|
| 3.6% (standard) | 70.1 ± 0.8 | 68.1 ± 1.3 | 72.2 ± 0.9 | 66.2 ± 1.9 | 68.7 ± 1.0 | **74.2 ± 0.3** |
| 1% | 59.4 ± 5.4 | 58.3 ± 4.0 | 46.5 ± 9.3 | 61.3 ± 3.9 | 63.3 ± 1.8 | **67.4 ± 2.3** |
| 0.5% | 45.3 ± 6.6 | 47.7 ± 4.4 | 38.2 ± 7.1 | 53.2 ± 4.0 | 53.8 ± 4.7 | **55.1 ± 3.2** |
| 0.3% | 39.3 ± 4.9 | 39.2 ± 6.3 | 30.9 ± 6.9 | 44.4 ± 4.5 | 46.7 ± 5.6 | **48.3 ± 3.5** |

Table 7: Accuracy (%) averaged over 10 runs on the Citeseer dataset.

| Training Split | Chebyshev | GCN | GAT | LNet | AdaLNet | DFNet |
|---|---|---|---|---|---|---|
| 0.3% (standard) | 69.8 ± 1.1 | 77.8 ± 0.7 | 76.7 ± 0.5 | 78.3 ± 0.3 | 78.1 ± 0.4 | **84.3 ± 0.4** |
| 0.1% | 55.2 ± 6.8 | 73.0 ± 5.5 | 59.6 ± 9.5 | 73.4 ± 5.1 | 72.8 ± 4.6 | **75.2 ± 3.6** |
| 0.05% | 48.2 ± 7.4 | 64.6 ± 7.5 | 50.4 ± 9.7 | 68.8 ± 5.6 | 66.0 ± 4.5 | **67.2 ± 7.3** |
| 0.03% | 45.3 ± 4.5 | 57.9 ± 8.1 | 50.9 ± 8.8 | 60.4 ± 8.6 | **61.0 ± 8.7** | 59.3 ± 6.6 |

Table 8: Accuracy (%) averaged over 10 runs on the Pubmed dataset.

Tables 6-8 present the experimental results. Table 6 shows that DFNet performs significantly better than all the other models over the Cora dataset, including LNet and AdaLNet proposed in [23]. Similarly, Table 7 shows that DFNet performs significantly better than all the other models over the Citeseer dataset. For the Pubmed dataset, as shown in Table 8, DFNet performs significantly better than almost all the other models, except for only one case in which DFNet performs slightly worse than AdaLNet using the splitting 0.03%. These results demonstrate the robustness of our model DFNet.

