[Supplementary Material · NeurIPS_2019_DFNets_Poster.pdf]

# DFNets: Spectral CNNs for Graphs with Feedback-Looped Filters

Asiri Wijesinghe and Qing Wang

Research School of Computer Science, ANU College of Engineering and Computer Science
The Australian National University, Canberra ACT 0200, Australia
{asiri.wijesinghe, qing.wang}@anu.edu.au

## Introduction

- We propose *Distributed Feedback-Looped Network* (DFNet) which is a novel spectral CNN architecture with feedback-looped graph filters.

## Feedback-Looped Filters

- Feedback-looped filters belong to a class of Auto Regressive Moving Average (ARMA) filters.

$$h_{\psi,\phi}(L)x = \left(I + \sum_{j=1}^{p} \psi_j L^j\right)^{-1}\left(\sum_{j=0}^{q} \phi_j L^j\right)x, \qquad (1)$$

where $p$ and $q$ refer to the *feedback* and *feedforward* degrees, respectively. $\psi \in \mathbb{C}^p$ and $\phi \in \mathbb{C}^{q+1}$ are two vectors of complex coefficients.

- The frequency response of feedback-looped filters is defined as:

$$h(\lambda_i) = \frac{\sum_{j=0}^{q} \phi_j \lambda_i^j}{1 + \sum_{j=1}^{p} \psi_j \lambda_i^j}. \qquad (2)$$

- To circumvent the issue of matrix inversion for large graphs, *feedback-looped filters* use the following approximation:

$$\bar{x}^{(0)} = x \text{ and } \bar{x}^{(t)} = -\sum_{j=1}^{p} \psi_j \tilde{L}^j \bar{x}^{(t-1)} + \sum_{j=0}^{q} \phi_j \tilde{L}^j x, \qquad (3)$$

where $\tilde{L} = \hat{L} - (\frac{\hat{\lambda}_{max}}{2})I$, $\hat{L} = I - \hat{D}^{-1/2}\hat{A}\hat{D}^{-1/2}$, $\hat{A} = A + I$, $\hat{D}_{ii} = \Sigma_j \hat{A}_{ij}$ and $\hat{\lambda}_{max}$ is the largest eigenvalue of $\hat{L}$.

- To alleviate the issues of gradient vanishing/ exploding and numerical instabilities, we use two techniques:

  - **Scaled-normalization technique:** centralizes the eigenvalues of the Laplacian $\hat{L}$ and reduces its spectral radius bound.

  - **Cut-off frequency technique:** allows the generation of ideal high-pass filters so as to sharpen a signal by amplifying its graph Fourier coefficients.

## Coefficient Optimization

- We aim to find the optimal coefficients $\psi$ and $\phi$ that make the frequency response as close as possible to the desired frequency response,

$$\acute{e}(\tilde{\lambda}_i) = \hat{h}(\tilde{\lambda}_i) - \frac{\sum_{j=0}^{q} \phi_j \tilde{\lambda}_i^j}{1 + \sum_{j=1}^{p} \psi_j \tilde{\lambda}_i^j} \qquad (4)$$

- Linear approximation of the error (w.r.t. $\psi$ and $\phi$) is defined as:

$$e(\tilde{\lambda}_i) = \hat{h}(\tilde{\lambda}_i) + \hat{h}(\tilde{\lambda}_i) \sum_{j=1}^{p} \psi_j \tilde{\lambda}_i^j - \sum_{j=0}^{q} \phi_j \tilde{\lambda}_i^j. \qquad (5)$$

- Let $\alpha \in \mathbb{R}^{n \times p}$ with $\alpha_{ij} = \tilde{\lambda}_i^j$ and $\beta \in \mathbb{R}^{n \times (q+1)}$ with $\beta_{ij} = \tilde{\lambda}_i^{j-1}$ be two Vandermonde-like matrices. The coefficients $\psi$ and $\phi$ can be learned by minimizing $e$ as a convex constrained least-squares optimization problem:

$$\textbf{minimize}_{\psi,\phi} \; ||\hat{h} + diag(\hat{h})\alpha\psi - \beta\phi||_2 \qquad (6)$$
$$\textbf{subject to} \; ||\alpha\psi||_\infty \leq \gamma \text{ and } \gamma < 1$$

## Spectral Convolutional Layer

- Let $\mathbf{P} = -\sum_{j=1}^{p} \psi_j \tilde{L}^j$ and $\mathbf{Q} = \sum_{j=0}^{q} \phi_j \tilde{L}^j$. The propagation rule of a spectral convolutional layer is defined as:

$$\bar{X}^{(t+1)} = \sigma(\mathbf{P}\bar{X}^{(t)}\theta_1^{(t)} + \mathbf{Q}X\theta_2^{(t)} + \mu(\theta_1^{(t)}; \theta_2^{(t)}) + b), \qquad (7)$$

where $\sigma$ refers to a non-linear activation function and $\bar{X}^{(0)} = X \in \mathbb{R}^{n \times f}$. $\bar{X}^{(t)}$ is a matrix of activations in the $t^{th}$ layer.

## Theoretical Analysis

- DFNets has several nice properties:

  - Improved localization
  - Efficient computation
  - Guaranteed stability
  - Linear convergence
  - Universal design
  - Dense architecture

| Spectral Graph Filter | Type | Learning Complexity | Time Complexity | Memory Complexity |
|---|---|---|---|---|
| Chebyshev filters | Polynomial | $O(k)$ | $O(km)$ | $O(m)$ |
| Lanczos filters | | $O(k)$ | $O(km^2)$ | $O(m^2)$ |
| Cayley filters | | $O((r+1)k)$ | $O((r+1)km)$ | $O(m)$ |
| ARMA$_1$ filters | Rational polynomial | $O(t)$ | $O(tm)$ | $O(m)$ |
| $d$ parallel ARMA$_1$ filters | | $O(t)$ | $O(tm)$ | $O(dm)$ |
| Feedback-looped filters (ours) | | $O(tp+q)$ | $O((tp+q)m)$ | $O(m)$ |

## Numerical Experiments

- Comparison with the state-of-the-art methods.

| Model | Cora | Citeseer | Pubmed | NELL |
|---|---|---|---|---|
| SemiEmb | 59.0 | 59.6 | 71.1 | 26.7 |
| LP | 68.0 | 45.3 | 63.0 | 26.5 |
| DeepWalk | 67.2 | 43.2 | 65.3 | 58.1 |
| ICA | 75.1 | 69.1 | 73.9 | 23.1 |
| Planetoid* | 64.7 | 75.7 | 77.2 | 61.9 |
| Chebyshev | 81.2 | 69.8 | 74.4 | - |
| GCN | 81.5 | 70.3 | 79.0 | 66.0 |
| LNet | 79.5 | 66.2 | 78.3 | - |
| AdaLNet | 80.4 | 68.7 | 78.1 | - |
| CayleyNet | 81.9* | - | - | - |
| ARMA$_1$ | 83.4 | 72.5 | 78.9 | - |
| GAT | 83.0 | 72.5 | 79.0 | - |
| GCN + DenseBlock | 82.7 ± 0.5 | 71.3 ± 0.3 | 81.5 ± 0.5 | 66.4 ± 0.3 |
| GAT + Dense Block | 83.8 ± 0.3 | 73.1 ± 0.3 | 81.8 ± 0.3 | - |
| DFNet (ours) | **85.2 ± 0.5** | 74.2 ± 0.3 | 84.3 ± 0.4 | 68.3 ± 0.4 |
| DFNet-ATT (ours) | **86.0 ± 0.4** | 74.7 ± 0.4 | 85.2 ± 0.3 | 68.8 ± 0.3 |
| DF-ATT (ours) | 83.4 ± 0.5 | 73.1 ± 0.4 | **82.3 ± 0.3** | 67.6 ± 0.3 |

- Comparison under different polynomial orders (DFNet).

- Node embeddings (top: Pubmed ; bottom: Cora).

(a) GCN (b) GAT (c) DFNet (ours)

(a) GCN (b) GAT (c) DFNet (ours)