[Reviews · NeurIPS 2019]

Reviewer 1



The paper presents a spectral convolutional layer using feedback-looped filters and show how it can be used efficiently by means of scaled normalization and cut-off frequency. It is clearly written and easy to follow. Introduction and walk-through on spectral convolutions is good and motivates the work. It would be nice to see how spectral methods compare to non-spectral ones, and if there is some interesting aspect that should be considered. The method section could take more space to elaborate on 3.2 as it is key for the method as in this current form I am not sure it is sufficiently detailed and Alternatively details could be provided in the experimental setup section, as long as they are sufficient to grasp details of the proposed method. Section 3.2 contains some details on initialization that should go in experiments, unless they are needed to explain something about the method. If so then please elaborate and explain better. No ablation study has been performed with respect to the architectural choice of dense net. What is the difference in performance? Hope much would other methods, such as GCN, gain from just this architecture? Section 4.5 should define better what are criteria for success, are dense clusters better than sparse ones for example? Figure 4 and 5 are not sufficiently explained to say anything relevant about the methods in my opinion. Please elaborate on this. In introduction Shuman, cite [30], was first to propose convolution on graphs via spectral decomposition. Overall I like the paper and the approach. Some polishing is still required and clarity should be improved.

Reviewer 2



In this paper, authors study the graph convolution networks. Especially, the authors propose a new filter to approximate the true Fourier transformation, in the same spirit of Chebyshev filters and others. The proposed filter is based on the ARMA filter of Eq. (6). This equation is computationally heavy, so the proposed feedback-looped filter approximates the equation as in Eq.(7). Then the actual implementation of Graph Neural Network is formulated as in Eq.(12). Experimental performance is measured with Cora/Citeseer/Pubmed scholarship networks and NELL knowledge graph. Accuracies in the semi-supervised node classification surpass the several standard GCN models. Since I am not familiar with signal processing literature, I have difficulty in understanding the technical contribution. There are a few barriers I encountered. (1) Why the rational polynomial coefficients are preferable for convolution filter development? Or, why the RPC is less sensitive in an underlying graph structure, compared to Chebyshev? (2) I read the ref. [16, 17], but cannot understand why the Eq.(7) serves as a (valid?) approximation for Eq.(6). Thus I cannot judge whether this is a technically reasonable, or interesting, approximation of the ARMA (Eq.(6)). In Eq. (12), a regularization term \mu is introduced for the layer construction. So, strictly speaking, the spectral convolution layer is not an exact implementation of the feedback-lopped filter, right? Then theoretical discussions in Sec 3.4 is not directly applicable for spectral convolution layers. Is this correct? Also, I expect an ablation study for the additional \mu terms to verify the effect of this regularizer on several experimental performances. I also have a concern about the experimental design. There is no description of how the authors tune the hyperparameters. In Figure 2, a pair of (p,q) = (5, 3) is the best for the "test" dataset, and the authors used this pair of (p, q) for the main result, table 4. Does this mean the authors tuned hyperparameters to maximize the test score? Or, the best (p,q) for the validation dataset is the same as the (p.q) for the test dataset? Please clarify how the hyperparameters are tuned during the training. + Experimental scores seem good. - It is difficult (at least for me) to fully understand the derivation of the feedback-looped filter. - Theoretical discussion is not for the actual convolutional layer (Eq.12), but its ideal(?) formulation (Eq.7) - The effect of \mu regularizer is not assessed. - Unclear hyperparameter optimization procedure. I really expect the author feedback to fully understand the contribution of the paper. ### After author feedback ### I'm happy to see the detailed author feedback. It definitely helps me understand the value of the submission more correctly. So I raised my review score upward. However, the manuscript is still somewhat difficult for some groups of readers. Further explanations and reference pointers will enlarge the potential audience.

Reviewer 3



Pros: 1, The proposed rational polynomial spectral filter is novel and interesting. 2, I like the summarization and the comparison of different spectral filters in terms of functional forms, complexity measures, etc. 3, The paper is clearly written. Cons: 1, I appreciate the authors' efforts in pushing forward the graph convolution by designing new spectral filters. However, from the current writing, I feel like there is a gap between the claimed new technique and practical benefits brought by the technique. For example, when you introduce and motivate the Cayley filters and the rational polynomial spectral filters in general, could you be more specific and provide more intuition on why “narrow frequency bands” is an issue practically, how these filters resolve this issue, and why improved localization would lead to better empirical performance. 2, I think an in-depth discussion and experimental comparison with the recently proposed multi-scale spectral graph convolution [23] is necessary since: (1) they use the neural networks as the learnable spectral filters which have better data-adaptive capacity; (2) they can cheaply compute the spectral convolution with very a large localization range. 3, In terms of experiments, I strongly discourage authors to only report performances on the small citation networks using the fix spit. As discussed in several recent works, e.g., [1], the ranking of different models dramatically changes when the split changes. I suggest authors either choose other larger datasets or report the performances over multiple random splits and/or decreasing the proportion of labeled nodes. 4, The experimental comparison is unfair to other methods. It is not surprising that adding dense connections improves the performance for many models. However, most of your baselines do not exploit this trick. I would suggest removing the dense connections and just compare the newly proposed spectral convolutional networks with the other methods. You need this experiment as an ablation study to clarify the core contribution. I did not see any empirical analysis on the claimed improved localization contribution. I suggest adding an ablation study to compare against other spectral filters, which have less controlled localization and same settings otherwise, e.g., Chebyshev filter. 5, What is the definition of learning complexity? [1] Shchur, Oleksandr, et al. "Pitfalls of graph neural network evaluation." arXiv preprint arXiv:1811.05868 (2018) ================================================================ Authors' rebuttal addressed most of my concerns on experimental comparison. One suggestion is that it would be more elegant and efficient to improve the coefficient optimization via amortized inference since it is costly to perform such an optimization per layer. I have changed the score accordingly.

[Author Response · NeurIPS 2019]

We would like to thank the reviewers for providing detailed and constructive reviews. Please find below our responses.

(1). **Why are the rational polynomial filters proposed in our work preferable for convolution operations? Why**
**can our proposed work resolve the issue "narrow frequency bands" and improve the localization?**

The rational polynomial filters proposed in our work have two key components: (i) *feed-forward filtering* which
performs the k-hop localization as polynomial filters; (ii) *feedback filtering* which filters out errors/noises in the output
of feed-forward filtering to improve the localization. The feedback filtering component can only be supported by our
proposed filters, not any other existing spectral filters. Moreover, unlike other rational polynomial filters, our filters
have guaranteed stability, i.e., rational polynomial coefficients are learned in a stable way as discussed in Section 3.4,
because the pole of our filters always lies in the unit circle of the z-plane.

"Narrow frequency bands" is an issue existing in the current spectral filters, including both polynomial and rational
polynomial filters (e.g. Chebyshev and Calyley), because they only accept a small band of frequencies of the Laplacian.
However, our proposed filters resolve this issue using a cut-off frequency technique, i.e., accepts frequencies higher
than a certain low cut-off frequency value and attenuates frequencies lower than that cut-off value. Thus, our proposed
filtrs can accept a wider range of frequencies of the Laplacian and capture better characteristic properties of a graph.

(2). **Why is Eq. (7) a valid approximation for Eq. (6)?**
An ARMA filter of Eq. (6) can filter a graph signal $x$ by altering its frequency response which has the form as presented
in Eq. (8). Then, according to Proposition 1 in [16], we also know that a feedback-looped filter using the approximation
in Eq. (7) has the same frequency response as described in Eq. (8) under a stability condition $||\alpha\psi||_\infty \leq \gamma$ and $\gamma < 1$.
This stability condition is required in Eq. (11) and discussed in Section 3.4 in detail.

(3). **Why is a regularization term $\mu$ introduced for the spectral convolution layer in Eq. (12)?**
The reason for introducing a regularization term $\mu$ is to alleviate the overfitting issue. Specifically, we use the unit-norm
constraint technique to restrict parameters of all layers in a small range and the kernel regularization technique to
penalize the parameters in each convolution layer during the training. In doing so, we can prevent the generation of
spurious features and thus improve accuracy of the prediction. This is our contribution on designing spectral convolution
layers suitable for the proposed filters. The theoretical discussion in Section 3.4 is still applicable in this case.

(4). **In the following, we clarify the questions relating to experiments. All the related results and the detailed**
**analysis will be provided in our final version and the supplementary materials of the paper.**

In Section 4.2, we have compared our proposed filters with self-attention, (i.e. DF-ATT) against GAT which uses
Chebyshev filters and self-attention. The results in Table 4 show that DF-ATT outperforms GAT over all four
datasets. Additionally, we have conducted experiments on comparing DFNet (our proposed filters+DenseBlock) with
GCN+DenseBlock and GAT+DenseBlock, as well as comparing our proposed filters with Chebyshev, GCN and Cayley.
The results below show that our proposed filters perform best, no matter whether the dense architecture is used.

| Model | Cora | Citeseer | Pubmed | NELL |
|---|---|---|---|---|
| GCN+DenseBlock | 82.7 ±0.5 | 71.3 ±0.3 | 81.5 ±0.5 | 66.4 ±0.3 |
| GAT+Dense Block | 83.8 ±0.3 | 73.1 ±0.3 | 81.8 ±0.3 | - |
| DFNet (ours) | **85.2 ±0.5** | **74.2 ±0.3** | **84.3 ±0.4** | **68.3 ±0.4** |

| Model | Cora | Citeseer | Pubmed |
|---|---|---|---|
| Chebyshev | 81.2 | 69.8 | 74.4 |
| GCN | 81.5 | 70.3 | 79.0 |
| Cayley | 81.9 | - | - |
| Feedback-looped (ours) | **82.6 ±0.3** | **71.5 ±0.4** | **81.7 ±0.6** |

For our models, hyperparameters were initially selected using the orthogonalization technique (a randomized search
strategy). Then, we used the validation dataset to select the best model. Thus, the best (p,q) for the validation dataset is
the same as the (p,q) used for the test dataset.

We have conducted experiments on our proposed filters with and without adding the $\mu$ term in Eq. (12). The table below
shows that, adding the $\mu$ term in Eq. (12) improves the performance on all four datasets.

| Model | Cora | Citeseer | Pubmed | NELL |
|---|---|---|---|---|
| Without adding the $\mu$ term in Eq. (12) | 84.2 ±0.3 | 73.1 ±0.4 | 83.1 ±0.3 | 67.4 ±0.4 |
| With adding the $\mu$ term in Eq. (12) | **85.2 ±0.5** | **74.2 ±0.3** | **84.3 ±0.4** | **68.3 ±0.4** |

We have benchmarked the performance of our DFNet model against the models in [23]. All experiments were repeated
10 times and the same hyperparameter settings in Section 4.2 were used for DFNet. The table below shows that DFNet
performs significantly better than all the models over the dataset Cora, including AdaLNet proposed in [23].

| Training Split | Chebyshev | GCN | GAT | LNet | AdaLNet | DFNet |
|---|---|---|---|---|---|---|
| 5.2% (standard split used in previous works [7, 19, 31]) | 78.0 ±1.2 | 80.5 ±0.8 | 82.6 ±0.7 | 79.5 ±1.8 | 80.4 ±1.1 | **85.2 ±0.5** |
| 3% (random split as in [23]) | 62.1 ±6.7 | 74.0 ±2.8 | 56.8 ±7.9 | 76.3 ±2.3 | 77.7 ±2.4 | **80.5 ±0.4** |
| 1% (random split as in [23] | 44.2 ±5.6 | 61.0 ±7.2 | 48.6 ±8.0 | 66.1 ±8.2 | 67.5 ±8.7 | **69.5 ±2.3** |
| 0.5% (random split as in [23] | 33.9 ±5.0 | 52.9 ±7.4 | 41.4 ±6.9 | 58.1 ±8.2 | 60.8 ±9.0 | **61.3 ±4.3** |

[Meta-Review · NeurIPS 2019]

This is a borderline paper. The consensus among reviewers is that the paper is appreciated and that the authors did a good job explaining the unclear parts of the paper.